# Experimenting Agriculture 4.0 with Sensors: A Data Fusion Approach between Remote Sensing, UAVs and Self-Driving Tractors [note 1]

**DOI:** 10.3390/s22207910

**Published:** 2022-10-18

**Authors:** Vincenzo Barrile, Silvia Simonetti, Rocco Citroni, Antonino Fotia, Giuliana Bilotta

**Affiliations:** 1DICEAM Department, University Mediterranea of Reggio Calabria, 89124 Reggio Calabria, Italy; 2Department of Engineering, Università degli Studi di Messina-Piazza Pugliatti, 1, 98122 Messina, Italy; 3Department of Electronic Engineering, University of Rome Tor Vergata, 00133 Roma, Italy

**Keywords:** vineyards, unmanned aerial vehicles, satellite imagery, agriculture 4.0, sensor networks

## Abstract

Geomatics is important for agriculture 4.0; in fact, it uses different types of data (remote sensing from satellites, Unmanned Aerial Vehicles-UAVs, GNSS, photogrammetry, laser scanners and other types of data) and therefore it uses data fusion techniques depending on the different applications to be carried out. This work aims to present on a study area concerning the integration of data acquired (using data fusion techniques) from remote sensing techniques, UAVs, autonomous driving machines and data fusion, all reprocessed and visualised in terms of results obtained through GIS (Geographic Information System). In this work we emphasize the importance of the integration of different methodologies and data fusion techniques, managing data of a different nature acquired with different methodologies to optimise vineyard cultivation and production. In particular, in this note we applied (focusing on a vineyard) geomatics-type methodologies developed in other works and integrated here to be used and optimised in order to make a contribution to agriculture 4.0. More specifically, we used the NDVI (Normalized Difference Vegetation Index) applied to multispectral satellite images and drone images (suitably combined) to identify the vigour of the plants. We then used an autonomous guided vehicle (equipped with sensors and monitoring systems) which, by estimating the optimal path, allows us to optimise fertilisation, irrigation, etc., by data fusion techniques using various types of sensors. Everything is visualised on a GIS to improve the management of the field according to its potential, also using historical data on the environmental, climatic and socioeconomic characteristics of the area. For this purpose, experiments of different types of Geomatics carried out individually on other application cases have been integrated into this work and are coordinated and integrated here in order to provide research/application cues for Agriculture 4.0.

## 1. Introduction

As the concept of digital transformation is making its way into all fields of daily life, revolutionizing the way we produce and interact, the applications of digital technologies tend to “specialize” in individual application sectors. Agriculture is often considered a “traditionalist” sector uninclined to changes; however, in recent years, it has benefited greatly from the technological evolution underway.

The term “industry 4.0” has been coined to indicate digital transformation in production environments; in the same vein, the entry of the technologies of the fourth industrial revolution into the agrifood sector can be called “agriculture 4.0”. Agriculture 4.0 is the result of the application of a series of innovative technologies in the agrifood field and it can be considered as an upgrade of precision agriculture. This was possible thanks to the automation of the collection and the integration and analysis of data collected directly from the fields through various types of sensors. In this context, digital technologies 4.0 are useful to support-thanks to data analysis-the farmer in his daily activity and in planning strategies for his business, including relationships with all links in the supply chain, generating a virtuous circle able to create value for the individual company and in cascade for its partners. Thanks to these new solutions and the application of digital technologies, from the IoT to artificial intelligence, from the analysis of large amounts of data to self-driving tractors to the use of drones, farms can increase profitability and the economic, environmental and social sustainability of its business. The beginning of the application of technologies for precision agriculture in Italy dates to the 1990s; basically, it involves using digital solutions for specific interventions, which take into account in particular the needs of the soil and plants. The aims of these interventions are to improve the production yield of the plantations as much as possible and contain costs and environmental impact. This category includes, for example, all interventions to make irrigation more efficient without wasting water resources or causing the plants to suffer, planting technologies adapted to the biochemical and physical characteristics of the soil on which the intervention is carried out, the administration of pesticides commensurate to the specific needs of each area and plant, or of fertilizers only in the necessary quantity and at the most useful times.

For this reason, precision agriculture, in addition to being the predecessor of agriculture 4.0, is also one of the cornerstones of the latter, because it lays the foundations for adapting production processes to individual needs thanks to targeted and timely interventions. All these interventions can adapt to the needs of the moment (through GIS, different types of sensor data, and the use of data fusion techniques, production peculiarities can also be estimated). The basis for making these technologies more effective is the real-time use of data coming from the fields. Thanks to sensors able transmit information, installed on fields or agricultural machinery, it will be possible to make timely and effective decisions, which can also be entrusted to automated systems. In general, the main advantages of agriculture 4.0 are those, as we said, of a rationalization of resources, with a positive economic impact for the companies in the supply chain. A path of products-from field to table-aimed at maximizing sustainability also has a positive impact on health, since it will be possible to bring better controlled and fresher products to final consumers than with traditional techniques. To quantify these advantages, there is talk of a saving of around 30% for production inputs and a 20% increase in productivity, with limited use of chemicals. Then focusing on the use of data, it must be added that being able to count on the real-time analysis of the information coming from the fields is extremely useful to manage any activity related to agriculture in a faster and, therefore, more efficient way. In fact, thanks to the data analysis, it will be possible to make the use of agricultural machinery as efficient as possible, or to use only the amount of water needed, without waste. Thanks to the same set of information, it will also be possible to prevent plant diseases or counteract pests, limiting damage when problems arise thanks to constant and simultaneous monitoring of crops. Moreover, it should be emphasized that these are advantages that can be obtained regardless of the type of crop.

This study starts from work previously carried out on a specific agricultural area, with the aim of reanalysis with different tools and techniques in order to find a more efficient monitoring solution.

It is possible to use Geomatics techniques, and thus to use satellite images, multispectral drone images (and there are already numerous analyses on the integration of satellite images and drone images to improve image quality and productivity), other sensors (humidity, pressure, wind, temperature), merge these data with data fusion techniques, manage the use of automated vehicles, and collect everything using GIS to optimise the agricultural production process (fertilisation, irrigation, etc....) according to the needs of the population.

The practice of Precision Agriculture (PA) and, recently, Agriculture 4.0, has garnered a lot of attention in recent years. Through the integration of information technology and agronomic practices, it has become possible to automate the management of parcels of land [1].

The literature cited below highlights the research questions and useful information and underlines the shortcomings of previous studies.

Precision agriculture is a management strategy [2] that utilizes information technology to collect data from multiple sources in order to use them in decisions regarding field production activities [3]. The goal of this strategy is to integrate the ideas of business management and process automation. Agriculture 4.0 additionally brings together various innovative methodologies applied from time to time in other sectors, such as the identification of optimal routes for self-driving tractors.

Crop monitoring, which is based on observations carried out directly on crops in place in order to obtain data on phenological stages, nutritional status [4,5,6], phytosanitary status, production expectations [7] and production maps [8], is of particular interest in order to accomplish this. The monitoring of crops relies on observations made directly on the crops in their natural environment. Since massive amounts of data need to be gathered and processed, process automation is necessary [9].

The monitoring of crops makes use of remote sensing data and is predicated on the link that exists between several parameters relating to the leaf curtain [10]. These parameters can express the vegetative–productive responses of plants and evaluate the variability as a function of the different behaviours of surfaces and bodies [11] to the phenomenon of absorption or reflection of light in the visible and infrared regions [12].

Large agricultural regions have been monitored using satellite remote sensing ever since the 1970s for the purpose of stock forecasting [13], which has resulted in the provision of useful data for the industry of agriculture itself. The unique optical behaviour of plants in the infrared radiation band makes remote sensing techniques particularly useful for evaluating vegetative health [14], as these techniques are useful in practice [15]. The time-consuming and financially burdensome flights of airplanes fitted with specialized cameras were quickly replaced by satellites that, while continuously orbiting the Earth, acquire data on the electromagnetic emission of objects on the Earth’s surface, and consequently also of the crops, with their multispectral sensors, if passive, or radar, if active. This has resulted in a significant reduction in the cost of collecting this information. However, passive sensors have limitations; acquisition is necessarily diurnal and hindered by any cloud cover. Furthermore, the level of detail that is obtainable precludes performing particular kinds of analyses on smaller parcels of land.

On the other hand, unmanned aerial vehicles (UAVs) have the potential to be very helpful because they can collect more specific georeferenced information using a variety of sensors [16,17,18,19].

In viticulture in particular, optimising vineyard cultivation and yield procedures through the use of automatic cultivation machines and data fusion, which faces challenges during production cycles by defining an adequate crop management, the Agriculture 4.0 approach has as its ultimate goal the improvement of vineyard yield and grape quality while simultaneously reducing all wastes, costs, and the negative impact on the environment [20].

The information collected by optical sensors in multispectral and hyperspectral imaging systems is utilized in the calculation of a diverse range of indices related to crop production (such as the Leaf Area Index (LAI) [15,21]). The normalized difference vegetation index (NDVI) is one of the indices that is utilised the most frequently because of its relationship to crop vigour and, as a result of this relationship, to the estimated quantity and quality of field production.

The MultiSpectral Instrument (MSI) of Sentinel 2 covers large areas, and many satellite programs (i.e., Landsat, Sentinel-1 and Sentinel-2) now freely supply datasets, which promotes the exploitation of satellite imagery for many applications, including agricultural applications, as multi-sensor and multiresolution data fusion [22,23,24,25,26]. The Sentinel-2 satellite, which was developed by the European Space Agency (ESA), has a resolution of one decametre, a revisitation time of six days, and an efficient resolution for analysing crop variability and conditions. If, on the other hand, we consider crops to be more like orchards and vineyards (with breaks in their layouts), then remote sensing becomes more challenging. Actually, the occurrence of paths between yields and weedy vegetation within the cultivated land can have a noticeable impact on the overall calculation of spectral indices, which in turn leads to a less precise evaluation of the crop’s status. New methods and algorithms were developed that use multispectral information from UAVs for circumventing this criticality [27]. These advancements have been made in order to resolve this critical issue.

Low-altitude platforms, as Unmanned Aerial Vehicles (UAVs) with airborne sensors, can differentiate pure canopy pixels from other objects by acquiring images with a high resolution and having flexible flight planning [28]. This allows for the classification of details within canopies.

In particular, it is possible to successfully combine an unmanned rotary-wing platform with a multispectral sensor in order to detect and monitor water-stressed areas of orchards, vineyards, and olive groves. This is possible thanks to the fact that the two technologies can be successfully combined.

In PA, the NDVI index is a parameter that is used because it is directly related to the health of the vegetation. This allows problems such as a lack of nutrients, the presence of parasitic infections, or conditions of water stress to be discovered. The NDVI index is calculated by processing images that were taken in the infrared. The early detection of such situations enables intervention that is both targeted and effective, which results in cost savings and increased crop yield. Infrared detection is frequently capable of identifying issues well in advance of their becoming obvious to the human eye [29,30].

In this article we present, among other things, a comprehensive vineyard survey that also compares MultiSpectral Instrument (MSI) data from a satellite with decametre resolution and from a low-altitude UAV platform. This allowed us to better understand the differences between the two types of instruments. The performance of Sentinel-2′s MSI, WorldVew and the aerial UAV sensors, both with very high resolution, when considering the relationship between crop vigour and NDVI was determined. In order to investigate the role played by the various vineyard components, satellite data were compared with UAV images using the following three NDVI indices [31,32]: (i) the entire agricultural area; (ii) only the vine canopies; and (iii) only the inter-row soil. These indices were calculated by comparing the UAV data with satellite images.

The multispectral sensors used on UAVs are capable of recording at least three channels, as a regular camera would, but one of those channels is replaced by infrared. Although multispectral sensors can acquire information in more than four bands and multispectral cameras are capable of recording more than the three channels defined here, for the purposes of this application, each image will consist of two visible colours in addition to infrared [33,34,35,36]. Because of this, the NDVI index can be calculated from a single image using a modified version of the conventional formula. The processing is handled in an automated practice by the GIS which we used (QGis). The maps that are obtained after the processing are false-colour maps known as “Vigour Maps” [37,38]. On these maps, red represents regions that have the highest possible vitality [39].

Recent literature also exists on data fusion techniques for agriculture: on input devices synchronised with microcontrollers and sending data from sensors via IoT (Internet of Things) devices to the cloud [40] and the challenges and complexity of Agriculture 4.0 [41].

It is obvious that the methodology that has been proposed can include other kinds of crops that are grown in rows, with crop canopies that do not extend over the entire area of cultivation, or where there is a significant presence of bare soil or grass [42,43,44].

## 2. Materials and Methods

### 2.1. Remote Sensing

Remote sensing is the acquisition of information about an object or phenomenon without coming into physical contact with the object. In this case we will refer to Remote Sensing from Earth Observation by satellite and, although it is used in numerous fields, in our case it is used for monitoring purposes in agriculture, particularly vineyard cultivation.

It is crucial for winemakers to gain an accurate understanding of the spatial variability both between and within crops to be able to make accurate predictions regarding yield and quality. The Normalized Difference Vegetation Index (NDVI) is one of the most widely used indices because it is related to crop vigour and, as a result, to estimated quantity and quality of field production.

Plants absorb solar radiation in the spectral region via photosynthetically active radiation (PAR), which they then use as an energy source in the photosynthesis process. Strong absorption at these wavelengths will only overheat the plant and potentially damage its tissue. As a result, plants appear relatively dark in the PAR spectrum and relatively bright in the near infrared spectrum. Clouds and snow, on the other hand, tend to be bright in the red band (as well as other visible wavelengths) and dark in the near infrared. Chlorophyll, a pigment found in leaves, strongly absorbs visible light for use in photosynthesis. In contrast, the cellular structure of leaves strongly reflects near-infrared light. The more leaves a plant has, the more wavelengths are affected, and thus the greater the amount of light involved. Because earth observation instruments collect data in the visible and near-infrared ranges, it was natural to use the large differences in reflectance of plants to determine their spatial distribution in satellite images.

The following formula is used to calculate the NDVI:(1)$$\mathrm{NDVI}=\frac{\left(\mathrm{NIR}-\mathrm{Red}\right)}{\left(\mathrm{NIR}+\mathrm{Red}\right)}$$

Red and NIR are abbreviations for spectral reflectance measurements obtained in the visible (red) and near-infrared regions, respectively.

### 2.2. UAVs/Sensors

Unmanned aerial vehicles (UAVs) are a type of robotic aircraft that are controlled by radio and have their own built-in control systems. They were initially developed in the 1920s for use in the military as a replacement for human pilots serving on hazardous missions. In the past, the disadvantages of high cost, large sensors, poor endurance, and primitive flight control systems caused civilian UAV use to develop slowly. At the beginning of the twenty-first century, only a few low-quality products were available for use in scientific research. These disadvantages still exist today. The market for low-cost unmanned aerial vehicles (UAVs) has expanded at a rapid rate thanks to the development of new technologies and the appearance of UAV manufacturers such as DJI (Shenzhen, China).

The successful transition of UAVs from military to civilian uses has been facilitated by the development of several different technologies. There is now an abundance of UAVs available to meet the demand in various fields of use, including scientific research.

The development of remote sensing technology has made it possible to devise a workable strategy for the collection of specific data used for mapping land-cover changes, monitoring drought conditions, and analysing complex characteristics across space and time. This technology uses a variety of sensors onboard satellites, airborne or unmanned aerial vehicles (UAVs), and it offers a variety of classification methods for vegetation at both large and small scales. A practical approach to designing strategies for the management of forest disasters can be found by employing the techniques of remote sensing. This can include evaluating landslide-prone areas through airborne, UAV, and ground-based remote sensing, as well as evaluating changes in vegetation cover after a wildfire for post-fire management by using satellite-based remote sensing and UAV.

There is technology available today that can automatically steer agricultural vehicles such as tractors [45,46,47] and harvesters along predefined paths using precise global navigation satellite systems (GNSS). Examples of these types of vehicles include tractors and combine harvesters. However, a human operator is still required in order to monitor the surrounding environment and take corrective action if any potential hazards come into view in front of the vehicle in order to guarantee its safe operation.

It is necessary for there to be no need for a human operator whatsoever for the autonomous farming vehicles to be able to operate in a manner that is both productive and risk-free without any assistance from a person. A safety system must be able to perform accurate obstacle detection and avoidance in real time while maintaining a high degree of reliability. Furthermore, in order to handle a wide variety of shifts in the illumination and weather conditions, multiple sensing modalities need to complement each other.

For a technological development of this magnitude, extensive research and experiments are required to investigate various sensor, detection algorithm, and fusion strategy combinations.

Today platforms such as drones support the integration of a wide variety of sensors employed for agriculture 4.0. However, the utilisation of these sensors in agriculture 4.0 is closely linked to their capacity to detect the signal over a greater spectral range. Fundamentally, in agricultural sensing technology four parameters must be analysed: the spectral, spatial, temporal, and radiometric resolutions. However, in most cases the sensors provide information based on their spectral resolution (multispectral, super-spectral and hyperspectral). Multispectral sensors typically use from 3 to 10 bands to cover the relevant spectrum. Early detection of the disease, improved irrigation, water management, faster and more accurate plant counts to optimize fertilizer application and pest control represent some advantages of this sensor. Super-spectral sensors use from 10 to 20 bands to cover broad portions of the spectrum. Hyperspectral sensors compared to multispectral sensors cover hundreds or thousands of narrower bands (10 to 20 nm), providing greater resolution and a highly detailed electromagnetic spectrum of agricultural fields. In addition, higher spatial resolution, ability to distinguish smaller elements, higher temporal resolution, higher radiometric sensitivity and the ability to detect small differences in radiated energy represent only some of the advantages.

### 2.3. Self-Driving Tractors/GIS/Other Sensors

Tractors used in agriculture are typically capable of working in any terrain. Moreover, the signals coming from the navigation sensors are subject to a great deal of unpredictability in terms of disturbances and noise sources. As a consequence of this, it is essential for the sensor fusion module to contain efficient methods for signal conditioning and estimating the state of the system.

When it comes to automated tractor guidance in the field, an accurate position measurement is absolutely necessary. Because the GPS antenna was mounted on the roof of the tractor cab, which was approximately three meters above the ground, any inclination of the tractor would result in an inaccurate position reading (roll and pitch). An architecture that uses edge devices to carry out a substantial amount of computation (edge computing), storage, and communication locally and routes it over the Internet backbone is called fog computing or fog networking, also known as fogging. A FOG was used to measure heading angle on this research platform. Utilizing Euclidean angles, a method of correction that compensates for positional errors caused by inclination-related factors was developed.

Hardware design and software design are the two components that make up the entirety of the overall structural design of unmanned agricultural machinery. The design of the hardware encompasses both the mechanical design and the circuit design. Programming for the control system execution process and algorithmic formulations for path tracking control are components of software design. By contrasting the traditional Proportional–Integral–Derivative (PID) control, fuzzy control, and fuzzy PID control, this article concludes that the fuzzy PID control algorithm should be used to control the steering of agricultural machinery, while the traditional incremental PID control algorithm should be used to control the speed of the vehicle body while it is in motion [48,49].

The quantity of sensors used to collect data in various settings, as well as the quality of the data collected by those sensors, has been steadily increasing. Even complex environments, such as agricultural areas, can now be “sensed” via a wide variety of equipment, which generates vast amounts of data that can be explored to provide helpful information about the area that is being observed. Examples of such environments include urban and wilderness areas. Because of this, an increased number of studies have been carried out in an effort to research the vast amounts of information that are hidden within the sensed data. However, it can be extremely difficult to transfer the advances made in experiments to the real-world conditions that are encountered in practice. There are two primary explanations for this phenomenon. To begin, the scope of the research projects that are described in scientific texts is typically restricted. This is due to the fact that the data that are utilized in these experiments typically do not cover all of the variables that are connected to the issue at hand. As a consequence of this, the results that are reported in those articles, despite the fact that they might appear to be encouraging, typically reveal nothing about the performance of the proposed technique under real-world conditions that are unrestricted. Second, even if the data adequately cover the variable conditions that are found in practice, the chosen sensing technology may not be able to acquire enough information to unambiguously resolve the data and provide enough information. This is a possibility even if the data adequately cover the variable conditions that are found in practice. For instance, even powerful artificial intelligence models that are fed with RGB digital images frequently fail to correctly identify plant diseases based on their symptoms. This is due to the fact that different disorders can produce visual signs that are similar to one another.

There are many sensors that can be used in Agriculture 4.0, for example, the Soil Moisture Sensor used in our case study, but also other environmental sensors, capable of providing data that can be used for cultivation decisions, also collected in time series that can therefore provide trends. Meteorological data [50] can also provide useful time series in agriculture.

As is well known, G.I.S. (Geographical Information System)/WebGIS is a tool for analysing, reporting and querying entities or events occurring in the territory. Particularly in Agriculture 4.0, the use of GIS allows researchers to integrate and manage data of different natures and, if properly implemented (open source), it also allows identification of optimal routes for vehicles and areas of greater interest in different areas if integrated with historical data.

The GIS makes use of images captured by UAVs as well as Very High-Resolution (VHR) satellite imagery categorized using OBIA. The Geographic Information System (GIS) is helpful for agriculture in general, and not just for the management of vineyards specifically. It takes into account the geomorphology of the land, as well as the climatic conditions (wind, rain, etc.), and the moisture conditions of the soil for the crops. This system is able to provide alerts in the event that interventions are required depending on the water stress experienced by the crop. As a result, we are able to highlight the optimal route for the tractor.

### 2.4. Data Fusion

Utilizing data fusion techniques is one approach to minimizing the gaps in coverage that are the result of insufficient data. The process of combining data from several different sources in order to produce information that is more precise, consistent, and concise than that which is provided by any individual data source is referred to as “data fusion.” There are also other definitions that are more stringent, which better fit specific contexts. Since the first half of the 1990s, people have been applying this method to solve agricultural problems, and recently, there has been an increase in the number of cases in which this method is used. Finding the most effective method to completely explore the synergy and complementarities that may exist between various kinds of data and sources of data is arguably the most difficult part of using techniques that involve the fusion of data. This is one of the main challenges that is involved in the use of data fusion techniques. This is especially the case when the data being compared have significantly different characteristics (for example, digital images and meteorological data). Given the wide variety of data sources and methods utilized in agricultural applications [40,41], it can be challenging to find a formalization for the data fusion process that is suitable for all of these applications. A perspective on the data fusion process is given here, broken down into three stages and applicable to the vast majority of situations. In the first, the corresponding attributes used for describing information in the various sources must be identified. This must be done before moving on to the next step. If the data sources are comparable, then finding such a correspondence is not difficult; however, if different types of data are used, then finding such a correspondence may be more difficult. This is one of the primary reasons that led to the development of the three distinct types of data fusion that are discussed in the paragraph that follows this one. In the second step, all of the distinct objects that are mentioned in the various data sources have to be located and arranged in the correct order. Because misalignments can lead to inconsistent representations and, as a result, unreliable answers, this step is particularly important when the data sources are images. Alignment errors are a common cause of these problems. The third step, which is the application of the actual data fusion, can be carried out once the data have been correctly identified and are consistent. In actual practice, addressing the inconsistent data that already exist is frequently ignored. Auxiliary tools, such as data profile techniques, which can reduce inconsistencies by extracting and exploring the metadata associated to the data being fused, have the potential to (at least partially) rectify this situation and bring it closer to an acceptable state.

We essentially perform data fusion on satellite images and drone images and then on various types of sensors using two different methodologies.

Geomatics uses various types of data (remote sensing from satellites, UAVs, and other data), so data fusion techniques are natural depending on the various applications to be carried out. This work aims to present on a study area the integration of remote sensing techniques, UAVs, autonomous driving machines, data fusion, and GIS in order to optimize the vineyard by optimizing cultivation and production by managing data of various types acquired with different methodologies. Geomatics-type methodologies used in other works and integrated here are specifically applied in this note for use and optimisation to contribute to agriculture 4.0.

## 3. Case Study

In the course of our research on a broader study area, focusing in particular on a vineyard that was located in Bova Superiore, a small municipality in the province of Reggio Calabria (South Italy), neighbourhood Briga, and that encompassed an area of approximately 2.2 hectares. The cultivated territory includes a series of parcels cultivated as vineyards, the most representative of which have, respectively, extensions of about 3.2 ha and 1.8 ha (Figure 1).

The vineyard is located on a sloping land with a varied morphology, with an alti-tude ranging from 600 to 800 m above sea level and an orientation mainly facing south.

The distance between rows is two meters, there is a gap of one metre between each row, and the width of the canopy along each row is approximately one metre. The planting took place in 2016 at the earliest.

There were differences in the vine’s vigour both within and between the plots that are likely to be found in the vineyard due to the irregular land morphology, such as soil characteristics and elevation.

### 3.1. Remote Sensing and UAV

In this study area, some experiments were conducted to test what we said above for agriculture 4.0. with particular reference to vineyards.

We conducted survey campaigns using satellites and drones between May and October of 2021 in order to extend the scope of the study to include different phenological phases of vines. Since the vigour does change over the course of the phenological cycle, we decided to acquire images at four different stages between flowering and ripening so that we could examine the plant in its various vegetative states. On the other hand, certain climatological patterns (such as below-average rainfall), which impeded the growth of plants, contributed to the stress that was experienced by the crops.

As satellite data were used, a Sentinel-2 Level 2A image was acquired on 24 May, 28 July, 27 August, and 21 September 2020 at 09:40 UTC and the image characteristics are reported in Table 1, and a WorldView-3 image acquired on 21 October 2021 (you can see an example of this in Figure 2).

Regarding instead the multispectral images obtained by drone, it is noted that a DJI Matrice 600 Pro drone [51] was used, integrating a multispectral sensor, Micasense Altum Camera [52] suitable for use in agriculture and with the ability to capture images of crops in both the visible spectrum and the infrared spectrum simultaneously. The following components are included in this system:A multispectral sensor recording crop images of crops in four spectral bands: Green (500 nm Bandwidth 40 nm), Red (660 nm Bandwidth 40 nm), Red-edge (735 nm Bandwidth 10 nm) and Near Infrared (790 nm Bandwidth 40 nm).An RGB camera (16 MP).An integrated 64 GB memory.A built-in brightness sensor (‘sunshine’ sensor) that records light situation and calibrates automatically the four multispectral sensors. The ‘sunshine’ sensor inte-grates an SD card slot to expand storage capacity.GPS and IMU (Inertial Measurement Unit).Table 2 shows UAV, sensor’s image and characteristics.

By carefully defining the sets of waypoints along the UAV route, it was possible to ensure that the aircraft would fly at a height of approximately 30 m above the ground. With these parameters, the aerial GSD images measure 5 cm (Table 2).

### 3.2. Self-Driving Vehicles/GIS

Regarding self-driving vehicles, an old experiment was adapted to simulate the behaviour of one or more tractors. The experiments we have conducted in the past concern self-driving vehicles intended for road monitoring, in Figure 3.

In this case, they are applied to agriculture monitoring.

The following items make up the standard instrumental equipment for such surveys:cameras that can capture images and movies and enable the acquisition and possibly later categorisation of objects, as well as integration with the mapping of the network technology in the area (water, electric, telephone, gas, etc.);Odometers and GPS.

The combination of uses will be determined by the kind of survey being conducted or the result that is to be obtained.

The use of Global Navigation Satellite System (GNSS) data and the transfer of information between the vehicle and the processing centre are essential components of any tracking system. In order to determine the position of the vehicle automatically, an experiment was carried out in advance to assess the efficacy of the various configurations; we selected the European Geostationary Navigation Overlay System (EGNOS) and the Real-Time Kinematic (RTK) method to check their respective performances.

The technological components of the system consist of a device for detecting the position (GPS), a transmission device (mobile phone), and a data processing centre equipped with a GIS platform. In addition to other information that is gleaned from active sensors on the vehicle, the data pertaining to the vehicle’s position and its instantaneous speed are transmitted from the vehicle to a processing point that is in charge of maintaining a database of field data.

Using a digital map of the area, special algorithms were applied to reduce errors, as positioning errors were present. These algorithms combine the position and trajectory of the vehicle as determined by the sensors with the routes that are available on the digital map. In the meantime, the information that is sent from the vehicle using the various sensors used enables an update to be made to the maps in terms of the routes to optimise the tractor’s path.

In order to determine the location of the vehicle, we analysed the results of both the EGNOS and RTK positioning systems and compared them. However, in order to calculate the position object, the RTK method requires real-time data processing, whereas the EGNOS system immediately provides location data. Although it requires more computational work, RTK provides more precise results than EGNOS does. When it comes to hardware instrumentation and software, the use of EGNOS is reliant on commercially available devices, whereas the RTK method necessitates the creation of customized software architecture.

In terms of communication systems, the possibility of using a Wi-Fi network offers benefits in terms of costs and speed as a result of the extremely low latency, but it also offers drawbacks in terms of the distance limits that can exist between antennas and the signal quality that can be achieved. In most cases, the maximum permissible distance between antennas is one hundred meters when the weather is clear, there is a direct line of sight between them, and there are no obstructions in the way. The signal quality can be affected by a variety of parameters, including the kind of antenna that is used and the possibility of interference.

The use of the mobile phone network, on the other hand, has a number of benefits, including complete independence among stations and vehicles, increased reliability due to the fact that it does not require compliance with minimum distances, and the capacity to process remote data remotely. The disadvantages include significantly higher latencies and increased costs. This is due to the fact that every device needs to be outfitted with a mobile network modulus and a SIM card that is associated with a particular data plan or a phone contract.

An open-source GIS/WebGIS was used for the processing and visualisation of the various data acquired. The GIS/WebGIS displays the results of the processing and the optimal routing of the routes, as well as highlighting the needs and requirements of the area, such as the need for irrigation timing, fertilisation timing, and anything else that may be useful for Agriculture 4.0, with appropriate alerts based on data fusion with other data.

In order to verify the effectiveness of the development, a platform for transmission to the GIS and user interface is created; it runs a procedure called Data Transfer GIS (DTGIS) for the subsequent export of the data acquired within the GIS, where the “historical” update is managed in the existing database.

In more detail, the DTGIS was designed to automatically transfer data acquired in three software modules, each with its own set of functions, to the GIS:

The Plug-in Module, which increases the number of recognizable and classifiable objects that can be represented;

The kernel, which interacts with users and coordinates the different modules, pre-processing and post-processing the Input/Output data of the modules themselves;

The GIS I/O (Input/Output) Module, which manages the interface with the GIS software.

In particular, the files (space database where the various attributes have been assigned to the objects) are given in Input in the GIS I/O module, returning Output polylines and polygons in shp-dbf format.

To implement the proposed system, a variety of algorithms and methodologies were used. Specifically, a multi-objective function based on Genetic Algorithms was used to determine the tractor’s route.

Furthermore, using Machine Learning algorithms, real-time hourly and continuous cycle trend information was obtained based on a comparison with recent and historical data (including the Backpropagation algorithm for the historical series).

IFTTT (If This Then That) is a programming language that allows for the real-time creation of condition chains called applets that are triggered by other services (e.g., Gmail, Facebook, Instagram, etc.) and can send a message when the user, for example, uses a hashtag in a tweet, or can send a copy of a Facebook photo to an archive when the user is tagged in it. IFTTT can automate processes related to home automation or web applications, such as receiving personalized weather forecasts or alerts in the event of an emergency, such as a flood. In our case, we used this service to send alerts and to automate tractor’s route when an alert is received.

Applet programming logic is of the following type: if a predetermined event occurs (trigger), then perform a predetermined action.

### 3.3. Other Types of Sensors

A wide range of available sensors can contribute significantly to agricultural practices. With the availability of low-cost data processing, solar panels, improved batteries and communications technology, the trend is now for these to operate wirelessly and transmit data to the user rather than relying on manual data collection. A variety of sensors are available for this purpose, including soil temperature, soil moisture content, air temperature and relative humidity, rainfall, solar radiation, barometric pressure, leaf wetness and wind speed and direction.

#### 3.3.1. Soil Moisture Sensor

As mentioned above, the ability to integrate several sensors is of crucial importance. For example, the Soil Moisture Sensor is used to measure volumetric moisture content of soils and other material for scientific research and agricultural applications. The sensor measures volumetric water content via the dielectric constant of the soil using capacitance technology. It uses a 70 MHz frequency, which minimizes salinity and textural effects, making it an ideal sensor in agricultural and standard scientific projects. Specifications on characteristics of the Soil Moisture Sensor are given in Table 3.

#### 3.3.2. Leaf Wetness Sensor

Most of these are based on well-known techniques and are used in other applications, but the leaf wetness sensor is aimed specifically at agricultural use and comprises a surface of conductive combs with a resistance of 2 MΩ when dry. This falls when condensation occurs on the surface, reaching approximately 5 kΩ when completely wet. The sensor generates a voltage that is inversely proportional to the degree of condensation.

#### 3.3.3. PH Sensor

The PH (Potential Hydrogen) meter is a device used to measure acidity and alkalinity levels in water, soil and photo chemicals. The PH meter consists of a voltmeter attached to a pH-responsive electrode varying in the range of 0 to 14.

The solutions with a pH value between 0 and 7 are acidic solutions with a large concentration of hydrogen ions, whereas solutions with a pH value between 8 and 14 have basic solutions with small concentrations of hydrogen. Solutions with a pH value of 7 are neutral solutions. In this process, we can detect the pH levels in the soil, in Table 4.

#### 3.3.4. Temperature and Humidity Sensor

The sensor has a humidity measuring module, a thermistor and an integrated circuit on the back of the sensor unit. The humidity measurement module consists of two electrodes. Sandwiched between the two electrodes is a substrate that is capable of holding moisture. Change in humidity alters the conductivity of the moisture-holding substrate, which at the same time changes the resistance. The integrated circuit then processes the change in the resistance and the humidity value is measured. On the other hand, a change in temperature changes the resistance of the thermistor, which is processed by the integrated circuit and the calibration results in a temperature value.

#### 3.3.5. Barometric Pressure Sensor

Barometric pressure sensors measure the absolute pressure of the air around them. This pressure varies with both the weather and altitude. Depending on how you interpret the data, you can monitor changes in the weather, measure altitude, or any other tasks that require an accurate pressure reading. The sensor consists of a piezoelectric transducer based on the characteristic of silicon to generate an electrical potential difference proportional to the mechanical stress applied on its surface. This type of transducer is characterized by extremely accurate performance and stable measurements of atmospheric pressure, with excellent repeatability and low hysteresis. An electronic amplifier circuit normalizes the output signal in the most common formats used by acquisition circuits (0–1 V, 4–20 mA). An electrical circuit for compensating the temperature allows more accurate measurements.

### 3.4. Data Fusion

As mentioned in Section 2.4, we use two different methodologies to perform data fusion on satellite and drone images, as well as data fusion on various types of sensors.

Sensors are used in agriculture for everything from weather monitoring to self-watering. Designers can create a prototype for a hardware environment to implement the data acquisition and mining process by using low-cost sensors. Thus, the relationship between sensors can be understood, and a sensor fusion test environment can be created. Various input devices are synchronized using a microcontroller system, and all data obtained from the sensors is wirelessly sent to the cloud by an IoT (Internet of Things) device, by recording and monitoring from the graphical user interface on the web as a real-time environment to apply data mining algorithms later. So, we obtain sensor data relations from various different data sources, such as soil moisture, but it is also possible to obtain data on light, temperature, humidity, rain, atmospheric pressure, air quality, and dew point. In the first experiment illustrated here, we use the soil moisture sensor. Each sensor data reading has a different effect on agricultural monitoring; however, reducing the number of sensors can reduce the cost of a system while still providing accurate observations via the proposed sensor substitution. A hardware test prototype, as well as a software design, are created for data monitoring and sensor fusion in various combinations.

Acquiring useful data from agricultural areas has always been difficult because they are often vast, remote, and vulnerable to weather events. Despite these obstacles, as technology advances and prices fall, a flood of new data is being collected. Although a wealth of data is being collected at various scales (e.g., proximal, aerial, satellite, and ancillary data), this has been geographically unequal, leaving some areas virtually devoid of useful data to help them face their specific challenges. However, even in areas with abundant resources and well-developed infrastructure, data and knowledge gaps persist, owing to the fact that agricultural environments are mostly uncontrolled and there are a plethora of factors that must be considered and properly measured in order to fully characterize a given area. As a result, even with very effective algorithms and a well-defined and limited-scope problem, data from a single sensor type are frequently unable to provide unambiguous answers. One possible solution that has been researched for decades is fusing the information contained in different sensors and data types. The concept behind data fusion is to investigate the complementarities and synergies of various types of data in order to extract more reliable and useful information about the areas being studied. While some success has been achieved, there are still many obstacles that prevent this type of approach from becoming more widely adopted. This is especially true in agricultural areas, which have highly complex environments.

Among the various data fusion methods, kriging was used, the weights of which were thought of as space variants and determined from calculations applied to plant growth phenomenology. In other words, rainfall and weather values in general, NDVI at varying seasons, and soil type were sampled at certain target points, from which a certain value of soil moisture was estimated. From these values, kriging was carried out at the location of the moisture sensor and compared with this truth value. The typical abatement parameters of the method are varied until a configuration of minima is found for which the value calculated by kriging and the value measured by the sensor are small. At this point, these abatement parameters can be used for kriging applicators to neighbouring areas, either from points observed by humidity sensors or derived from other devices.

So far, we have talked about data fusion between different sensors, but data fusion between images and sensors is also possible, as is the use of neural networks to improve image resolution.

The technology associated with the use of drones has undergone strong development in the last decade by improving the stability of the craft, lightening the structure, perfecting the precision and accuracy of acquisition and optimising the software for processing data. Among other things, this technology finds application in environmental monitoring, combining data acquisition over a wide area with high resolution and multispectral information. However, surveying with UAVs (Unmanned Aerial Vehicles) is not always cheaper than using satellite data. This is where the use of machine learning, and in particular, SuperResolution, comes in.

The freely available satellite data, as far as the Sentinel missions of the Copernicus programme are concerned, give a considerable advantage, but the resolution of these data may be too low for the studies to be carried out. It is therefore necessary to intervene with processing methods to improve the quality of the data. Furthermore, the timing of acquisition favours the use of satellite data over the drone survey and the processing of the related data because it is time-consuming. The satellite data, on the other hand, supplied already corrected in terms of reflectance, are directly usable after downloading.

With the use of a convolutional neural network, a procedure is applied that uses the satellite images as the basic data and allows a higher resolution product to be obtained. To achieve this, the VDSR (Very Deep Super Resolution) neural network is iplemented, using images acquired by drone for training the network. The aim of this work is to study the applicability of the VDSR (Very Deep Super Resolution) neural network in the context of remote sensing, using drone images as data.

Super-resolution, a process for obtaining high-resolution images from low-resolution images, compensates in Remote Sensing for limitations due to a spatial resolution that is not always adequately detailed. Single Image Super-Resolution (SIRS), in particular, aims to construct a high-resolution image from a single low-resolution image. A basic approach to achieve the improvement of an image’s resolution is interpolation, but there are other, more elaborate strategies that have the same goal.

The deep learning algorithm VDSR (Very-Deep Super-Resolution) is one of the possible techniques that can be used to perform the SISR process. Initially, the training of the neural network is necessary in order to then use the VDSR network to obtain a high-resolution image from a single low-resolution image. VDSR is a convolutional neural network CNN (Convolutional Neural Network) with the aim of relating high- and low-resolution images that differ mainly in high-frequency detail. The procedure is based on determining the residuals between the two images, i.e., a high-resolution reference image and a low-resolution image scaled to the same size as the reference image by means of bicubic interpolation.

The objective of the multispectral analysis was the calculation of the NDVI index, which can be obtained from the Red and NIR band.

The tests carried out on areas of different extension highlight the different possibility of using the image processed with the VDSR network as the survey area varies. For a portion of territory of the order of magnitude of one hundred metres, the data acquired with a drone possess resolution and detail that the other images cannot represent. For an area of approximately 25 hectares, the improvement obtained by processing with the VDSR neural network is enhanced; the extension is high enough to evaluate the use of the drone carefully, but not so large as to accept the detail of the satellite image. Here, the use of the neural network emphasises the edges of the framed elements more strongly, making them more easily recognisable. For an analysis area of the order of magnitude of several kilometres sideways or larger, processing with a VDSR neural network offers an improvement, but the detail required by the study can also be satisfied by using the original satellite image. A crucial aspect in the application of deep learning, which must be carefully evaluated in combination with the desired image enhancement, is the computing power required to perform the processing. The processing time for both the training of the neural network and its activation is non-negligible if adequate equipment is not available. To give an example, the training of the VDSR network used for the analyses in this study would have taken about ten days on an average commercial laptop with an Intel Core i5 5th Generation processor and 8-Gigabyte RAM. Furthermore, obtaining an image with a larger pixel size than the satellite image also increases the calculation time for subsequent processing, such as classification. The time factor negatively affects the evaluation of the practical use of the neural network, particularly when compared to other methods of improving the resolution of an image, such as interpolation.

## 4. Results

After selecting from the Sentinel and WorldView images and UAVs multispectral images, and other sensor (Soil Moisture Sensor) data, a data fusion procedure was performed with particular reference to areas A and B (Figure 4). So, a procedure has been put into place so that the value of the NDVI can be automatically determined from satellite images. In order to achieve homogenization of Sentinel and UAV data, it is necessary to automatically determine the value of NDVI derived from satellites (NDVIsat), through a downsampling of correlation between pixels s(i,j) from satellite and P(i,j) from UAV), to calculate the NDVI from UAV (NDVI_uav_) and to calculate both the NDVI for the leaf canopies of the vines (NDVI_vin_) and NDVI of inter-row area (NDVI_int_). In fact, an important tool for evaluating the variability in the vineyard and therefore the vines’ vigour is the NDVI index, thus calculated for the pixels of the Sentinel and WorldView image s(i,j) thanks to the spectral data) in RED and NIR bands:(2)$${\mathrm{NDVI}}_{\mathrm{sat}}\left(i,j\right)=\frac{n_{N}\left(i,j\right)-n_{R}\left(i,j\right)}{n_{N}\left(i,j\right)+n_{R}\left(i,j\right)}$$

A preliminary downsampling method of the high-resolution UAV images was used to allow the comparison of the UAV-based MSI and the satellite imaging. So, we proceeded to sampling the UAVs, data (at higher resolution) for comparing them with the corresponding satellite data, i.e., the set of UAV data D corresponding to P(i,j):(3)$$G\left(i,j\right)=\{d\left(u,v\right)\in D|\alpha_{s}\left(i,j+1\right)\leq \alpha_{d}\left(u,v\right)<\alpha_{s}\left(i,j\right),{\;\beta }_{s}\left(i,j\right)\leq \beta_{d}\left(u,v\right)<\beta_{s}\left(i+1,j\right),\;\forall u,v\}$$

Thus the satellite data s(i,j) and UAV data P(i,j) show the same subset of the vineyard. Three NDVIs were analysed from the VHR 2 data from the multispectral sensor mounted on the UAV, then compared with the satellite data on:(i) the entire cultivated area P(i,j):
(4)$${\mathrm{NDVI}}_{\mathrm{uav}}\left(i,j\right)=\frac{\sum_{u}\sum_{v}\frac{m_{N}\left(u,v\right)-m_{R}\left(u,v\right)}{m_{N}\left(u,v\right)+m_{R}\left(u,v\right)}}{\mathrm{card}\;P\left(i,j\right)}\forall d\left(u,v\right)\in P\left(i,j\right)$$

(ii) the pixels of the canopies:


(5)
$${\mathrm{NDVI}}_{\mathrm{vin}}\left(i,j\right)=\frac{\sum_{u}\sum_{v}\frac{m_{N}\left(u,v\right)-m_{R}\left(u,v\right)}{m_{N}\left(u,v\right)+m_{R}\left(u,v\right)}}{\mathrm{card}\;P\left(i,j\right)}\forall d\left(u,v\right)\in P_{\mathrm{vin}}\left(i,j\right)$$


(iii) the pixels of the inter-rows:


(6)
$${\mathrm{NDVI}}_{\mathrm{int}}\left(i,j\right)=\frac{\sum_{u}\sum_{v}\frac{m_{N}\left(u,v\right)-m_{R}\left(u,v\right)}{m_{N}\left(u,v\right)+m_{R}\left(u,v\right)}}{\mathrm{card}\;P\left(i,j\right)}\forall d\left(u,v\right)\in P_{\mathrm{int}}\left(i,j\right)$$


Table 5 shows the nomenclature of the symbols used.

In Figure 4 the results are shown.

Figure 4a shows the full set of pixels obtained from the NDVI_sat_ map, selected from Satellite imagery.

In Figure 4b an NDVI_uav_ map congruent (correctly aligned, at the same spatial resolution) to those derived from satellite imagery (NDVI_sat_) is shown.

In Figure 4c a complete NDVI_vin_ map is shown.

In Figure 4d the NDVI_int_ map of the inter-row ground, derived by processing the UAV images is shown.

The use of an image with a resolution of approximately 30 centimetres, such as a WorldView-3, would still enable a better definition of the vigour of the vines and, more generally, of the row crops. This is despite the fact that the resolution of the drone data is not comparable to that of the image.

The ground sampling distance (GSD) for the panchromatic band on WorldView-3 is 31 centimetres, while the GSD for the eight multispectral bands is 124 centimetres. Proceeding in the same manner as here with the UAV and using imagery obtained from the WorldView-3 satellite would result in an analysis of the vigour that is significantly more accurate. We could also provide a verification with Object-Based Image Analysis (OBIA), first using segmentation of the canopies and inter-row areas, then proceeding separately to the classification of the vigour through the various NDVIs found in the extraction of the objects formed with OBIA. Because the spaces between the rows are distinguishable (the data obtained from the decametric satellite sensor contribute to an inaccurate understanding of the actual vigour of the vines), we could also provide verifying (extracting objects directly from satellite imagery is one of the strengths of OBIA, which is used in a wide range of applications [22,24]).

Data fusion techniques make it possible to obtain complete information on an area and on the needs connected to cultivation from the fusion of different data [40,41] such as satellite data and UAV images, but also from different sensors such as soil moisture sensors (as Big Data [53]).

The automatic vehicle is useful as it is capable of working in any terrain including difficult terrain conditions, reducing human intervention.

Figure 5 depicts the optimized tractor path derived from data analysis, with the attention points for fertilization/irrigation derived from data fusion in green.

Finally, the GIS displays the results of the processing (Figure 6) and the optimal routing of the routes (Figure 5), and also highlights with appropriate alerts, depending on the data fusion with other data, the needs and requirements of the area, such as the need for irrigation timing, fertilisation timing and anything else that may be useful for Agriculture 4.0.

Furthermore, the use of historical data implemented in the GIS makes it possible to highlight areas where the analysis of historical and socio-economic data makes a different kind of cultivation appropriate (Figure 7).

Even though the method is still experimental, exploiting applications that have already been individually tested in other areas, these analyses nevertheless make it possible to clearly highlight what the contribution to Agriculture 4.0 can be from the integration of the various technologies of Geomatics. In particular, with the experiments carried out, it is possible to identify on a study area the optimal routes for tractors, the points where irrigation and top dressing are required, the areas that need intervention and the areas of vineyard vigour estimated through the use of the NDVI index with the pros and cons of the methodology.

## 5. Conclusions

Our article presents an introduction to a more in-depth analysis by comparing multispectral vineyard imagery obtained from satellite platforms such as Sentinel-2, at a resolution of ten meters, and ultra-high-resolution imagery acquired from WorldView satellite and low-altitude UAV platforms. Using NDVI as a measure of vineyard vitality, we compared the usefulness of the images obtained from the specified satellites and those obtained from UAVs. The chosen experimental site for the realization of four imaging campaigns that were scheduled according to the main phenological stages of the grapevine was a farm located in Bova Superiore, which is in the region of Calabria in Southern Italy.

As the aim of this work is to test methodologies for Agriculture 4.0, the activities conducted concerning data fusion methodologies on satellite images, UAV images, and additional sensors data as well as the use of a self-driving vehicle allow for experimentation in the area of Agriculture 4.0, leaving open broad research topics that can be worked on in the future.

In relation to the specific situation of the rows of vines, it is noted that new results can be obtained by changing sensors and with new, higher-resolution multispectral satellite images.

Past results have already shown that data acquired from decametre-resolution satellite systems (Sentinel-2) are insufficient to accurately assess vineyard conditions and crop variability. In fact, vineyard vigour may not agree with that of the inter-row zones, determining three distinct NDVI indices from the high-resolution UAV images, considering: (i) the entire cultivated area; (ii) only the vine canopy; and (iii) only the soil pixels between the rows. Indeed, the NDVI calculated from UAV images of only the pixels representing the vine canopy more accurately described the vigour of the vineyard. The proposed strategy can be applied to other types of crops that are cultivated with significant spaces between the rows.

The GIS that was developed for the purpose of monitoring and managing agricultural land with remote sensing using UAV images and VHR satellite imagery classified with OBIA is very helpful for agricultural management and produces alerts in the event that crop stress occurs.

This research is still open. Further experimentation will have to be carried out to optimise the system by making it usable and extracting more data to obtain final information to be further tested in the field or other areas to estimate the benefits of the method.

## Figures and Tables

**Figure 1 sensors-22-07910-f001:**
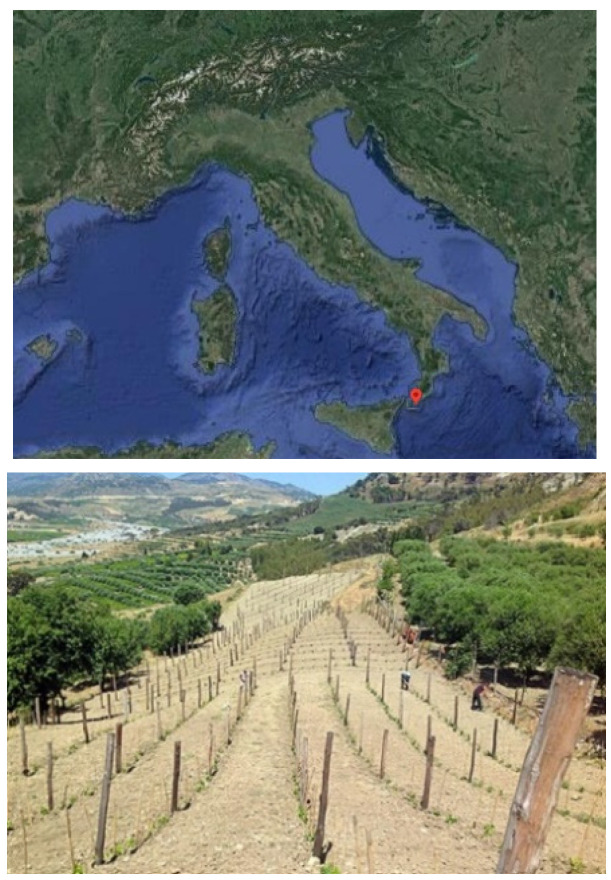
Study area: Bova Superiore in Calabria, Southern Italy.

**Figure 2 sensors-22-07910-f002:**
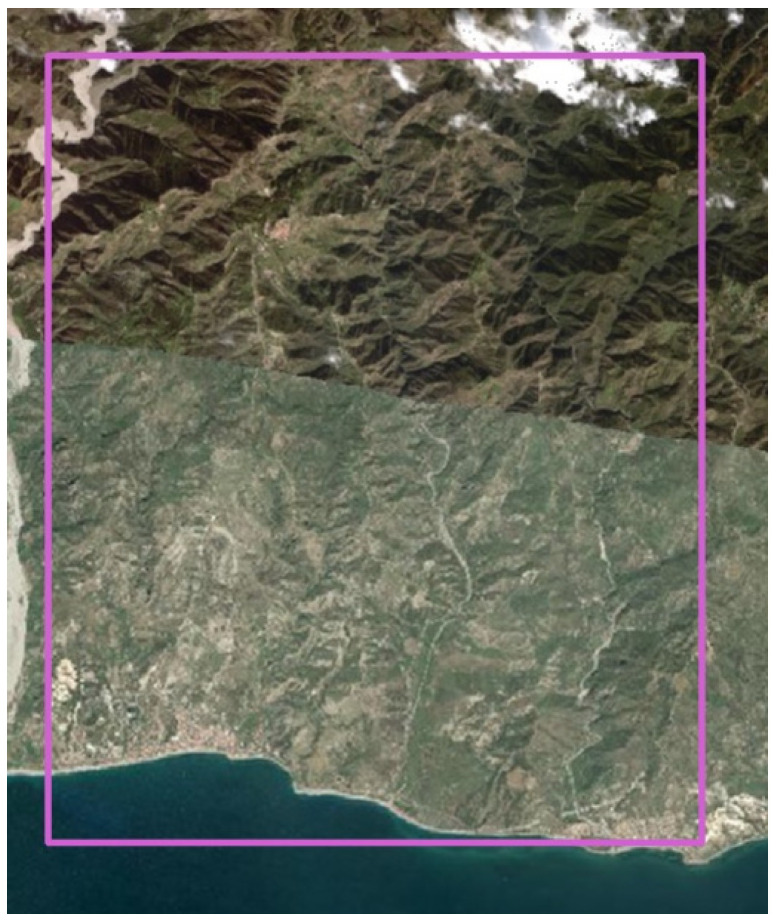
WorldView-3 acquired on 21 October 2021, resolution 30 cm, subset of the province of Reggio Calabria including the study area.

**Figure 3 sensors-22-07910-f003:**
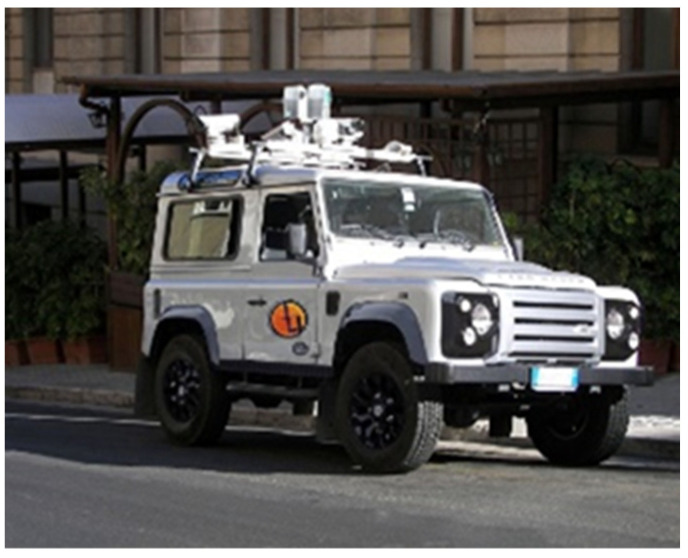
Equipped self-driving vehicle.

**Figure 4 sensors-22-07910-f004:**
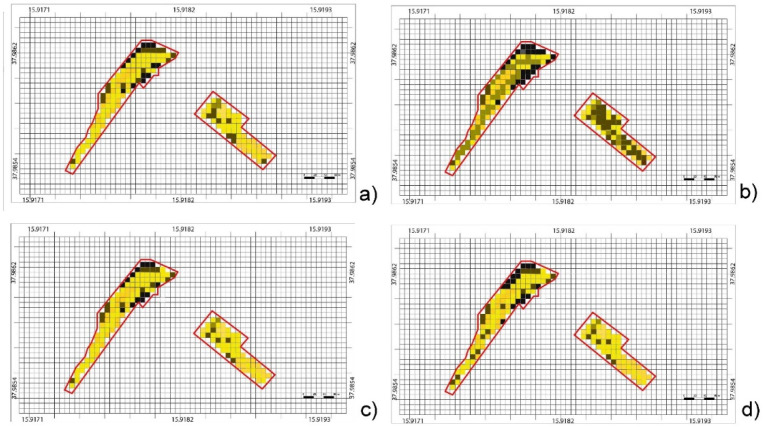
Complete NDVI (Normalized Difference Vegetation Index) maps: (**a**) NDVI_sat_ map, with pixels fully included in “Area A” and “Area B”, derived from satellite images S2, and (**b**) NDVI_uav_ obtained from UAV images D2. (**c**) Vineyard NDVI_vin_ map from UAV images D2 obtained only on canopy pixels Pvin, (**d**) NDVI_int_ map that considers inter-row ground P_int_.

**Figure 5 sensors-22-07910-f005:**
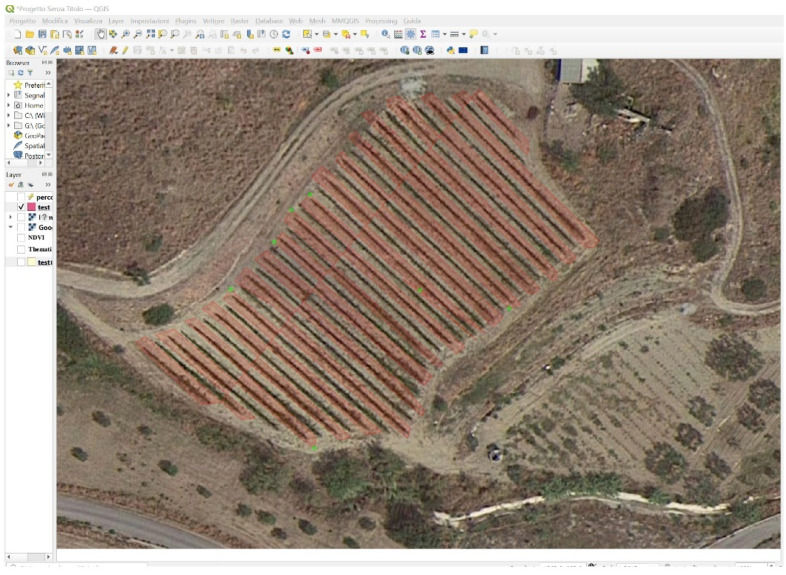
The optimised tractor path obtained from data analysis, with the attention points for fertilisation/irrigation obtained from data fusion shown in green.

**Figure 6 sensors-22-07910-f006:**
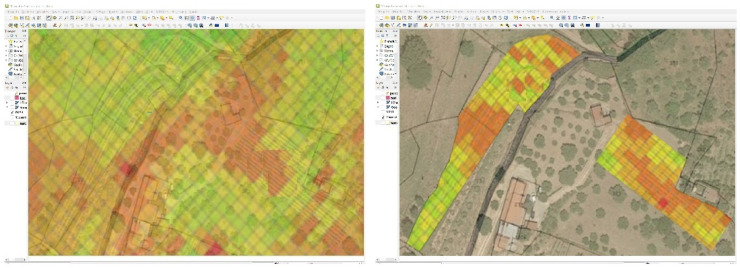
GIS (Geographic Information System), VHR (Very High Resolution) image: green = NDVI (Normalized Difference Vegetation Index) high; yellow = NDVI medium; red = NDVI low.

**Figure 7 sensors-22-07910-f007:**
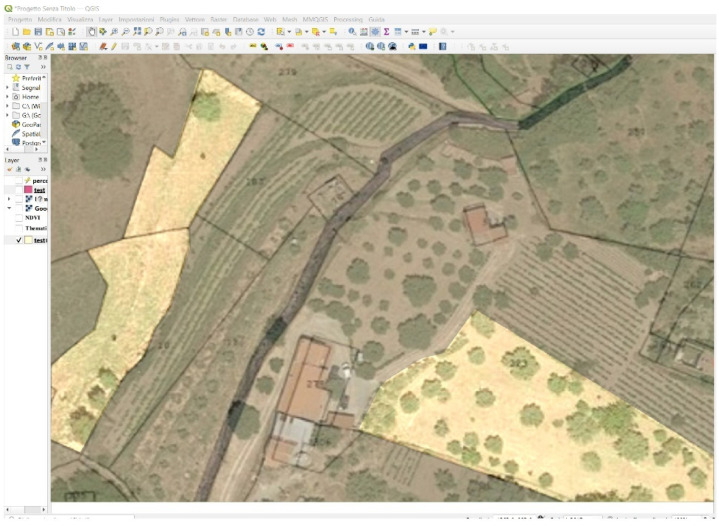
GIS (Geographic Information System): areas where the analysis of historical and socio-economic data makes a different kind of cultivation appropriate.

**Table 1 sensors-22-07910-t001:** Characteristics of satellite Sentinel 2 imagery.

Sentinel 2	
No. channels	13
Spectral bands used	B4-Red 650–680 nm B8-NIR 770–810 nm
Ground Sampling Distance (GSD) per band	10 m
Ground Dimension of the image	100 km × 100 km

**Table 2 sensors-22-07910-t002:** Platforms and sensor used: DJI Matrice 600 Pro and Micasense Altum Camera.

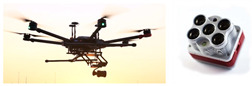	No. channels	4
	Spectral bands	B2–Red 640–680 mm B4–NIR 770–810 nm
	GSD per band	5.2 cm
	Flight speed	30 km/h
	Flight altitude	30 m
	FOV–Field-of-view	48° × 36.8°
	Ground Dimension of the Image	160 m × 30 m 100 m × 35 m

**Table 3 sensors-22-07910-t003:** Characteristics of the Soil Moisture Sensor.

Property	Characteristics
Accuracy	Apparent Dielectric Permittivity (εa): ±0.5 from εa of 2 to 10, ±2.5 from εa of 10 to 50 Soil Volumetric Water Content (VWC): Using standard calibration equation: ±0.03 m^3^/m^3^ (±3% VWC) typical in mineral soils that have solution electrical conductivity < 10 dS/m; using soil specific calibration, ±0.02 m^3^/m^3^ (±2% VWC) in any soil
Resolution	εa: 0.1 from εa of 1 to 30, 0.2 from εa of 30 to 50 VWC: 0.0008 m^3^/m^3^ (0.08% VWC) in mineral soils from 0 to 0.50 m^3^/m^3^ (0–50% VWC).
Range	εa: 1 (air) to 50 VWC: Calibration dependent; up to 0–57% VWC with polynomial equation
Sensor Type	Capacitance (frequency domain)
Dimensions	14.5 cm × 3.3 cm × 0.7 cm
5.5	Reduced soil microbial activity
Cable length	Sensors come standard with 5 m cable. Custom cable lengths available. Maximum cable length of 40 m. Please contact Decagon if you need longer cable lengths.
Measurement Time	10 ms
Power	3 VDC @ 12 mA–15 VDC @ 15 mA
Output	300–1250 mV, independent of excitation voltage
Operative Environment	Survival Temperature: −40–50 °C Operating Temperature: 0–50 °C
Connector Types	3.5 mm “stereo” plug or stripped and tinned lead wires

**Table 4 sensors-22-07910-t004:** PH (potential of Hydrogen) values and plants growth.

Soil pH	Plant Growth
>8.3	Too alkaline for most plants
7.5	Iron availability becomes a problem on alkaline soils.
7.2	6.8 to 7.2–near neutral 6.0 to 7.5–acceptable for most plants
7.0	
6.8	
5.5	Reduced soil microbial activity
<4.6	Too acidic for most plants

**Table 5 sensors-22-07910-t005:** Nomenclature.

Term	Nomenclature
d(u,v)	Pixel in row u and column v of D, raster matrix
D	High-resolution UAV multispectral image
P(i,j)	UAV pixels d(u,v) depicting the area of satellite pixels s(i,j)
P_vin_(i,j)	UAV pixels d(u,v) showing only vines canopy
P_int_(i,j)	UAV pixels d(u,v) depicting only inter-row ground
NDVI_sat_(i,j)	NDVI estimated using satellite images S
NDVI_uav_(i,j)	Entire NDVI calculated on UAV pixels in P(i,j)
NDVI_vin_(i,j)	NDVI calculated only on UAV pixels P_vin_(i,j) that represent the vine canopy
NDVI_int_(i,j)	NDVI calculated only on UAV pixels P_int_(i,j) showing inter-row ground
m_N_(i,j)	Reflectance values in the NIR band of pixels d(u,v)
m_R_(i,j)	DNs in the red band of pixels d(u,v)
n_N_(i,j)	DNs in the NIR band of pixels s(i,j)
n_R_(i,j)	DNs in the red band of pixels s(i,j)
s(i,j)	Pixels of row i and column j in the raster matrix S
S	Multispectral image 10 m resolution from Sentinel satellite
α_d_(u,v)	Latitude coordinate (in WGS84) of pixel d(u,v) centre
α_s_(i,j)	Latitude coordinate (in WGS84) of the upper left corner of pixel s(i,j)
β_d_(u,v)	Longitude coordinate (in WGS84) of pixel d(u,v) centre
β_s_(i,j)	Longitude coordinate (in WGS84) of the upper left corner of pixel s(i,j)

## Data Availability

Not applicable.

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
