# Peer review of "Experimenting Agriculture 4.0 with Sensors: A Data Fusion Approach between Remote Sensing, UAVs and Self-Driving Tractors"

_sensors, 2022, doi:10.3390/s22207910_

Round 1

Reviewer 1 Report

This is an interesting article, which presents a case study based on the application of agriculture 4.0 in the area of Calabria, Italy. It combines data collection techniques using sensor networks, satellite and drone imagery, data fusion and the generation of warnings and indicators, and even communication with tractors and harvesting machinery.

The approach is appropriate and is the main strength of the article, although some points for improvement have been identified and are detailed below. Publication is therefore recommended following these minor changes which I believe would improve the quality of the article.

Recomendations about content:

1.- The introduction (line 32) are differentiated the terms agriculture 4.0 and precision agriculture, implying and emphasising that these are two different concepts. However, in the rest of the article, the two terms seem to be used interchangeably. Thus, in the background section, the entire article is devoted to the concept of precision agriculture and the case is presented from this approach; and later, in section 2.2 (line 233) or when describing the case study (line 350), the term agriculture 4.0 is used. It is recommended in the introduction to clarify the difference between the two terms and to homogenise the term by one or the other when applied to the case study in question.

2.- In the paragraph associated with line 131, the NDVI index is introduced, of which some details and associated formulae are subsequently given. A shortcoming that has been noted in the article is that importance is given to this index, but no text area is dedicated to explaining in detail what it consists of and documenting its calculation in a more precise way. It is recommended that it be detailed so that the reader can find out how to calculate it or, failing that, provide references to which more information on it and its calculation can be obtained.

3.- In relation to the NDVI index, there is a moment on line 162 where it is written "This index can be found here", but no reference or associated linking information is provided.

4.- In lines 267 to 272 a lot of control techniques are mentioned with which it is indicated that they have been contrasted and, however, in the conclusions section it is only mentioned that a statistical analysis and/or analysis of variance has been carried out. If all these techniques mentioned in this paragraph have been applied and compared, it would be appreciated that in the conclusions section there is a space dedicated to an assessment and comparison of the results obtained with their application.

5.- In the paragraph on line 54, it is mentioned that the position from the GPS signal has been improved using the so-called Euclidean correction. It would be appreciated if details could be given on the degree of correction of the positioning error using this technique.

6.- In lines 334 and 335 the working area in the case study is georeferenced providing the coordinates based on latitude and longitude. However, the information is only provided in the range of minutes. It is recommended to correct and expand the information by giving the data based on X Degrees(o) Y Minutes (‘) Z seconds (“) indicating whether it is latitude North (N) or South (S), and the same for longitude, indicating whether it is East (E) or West (W). Thus, the recommended format would be, for example: latitude: 41° 56′ 54.3732" N, longitude: 87° 39′ 19.2024" W.

7.- I recommend giving some references of the "agricultural applications" indicated in line 310.

8.- In line 380, it would be advisable to include a reference to the manufacturer's website where the technical characteristics of the DJI Matrice 600 Pro Drone and the Micasense Altium camera are provided.

9.- Apart from what has already been commented on the comparison of the control techniques (PID, Fuzzy, etc) mentioned initially, the results also lack more description of the data fusion techniques used, which were indicated in the introduction that they were going to be used.

Recomendations about formatting:

1.- Both the numbering of some references and the way they are referenced should be revised. For example:

1.1.- On line 121, references 16-19 are placed and then skipped to 32. It is recommended that 32 be changed to 20. The same happens in other areas of the article. Something similar happens on lines 356 and 358 where reference 38 is moved to 44, which should be numbered as 39. It is recommended that the numbering order be revised and to ensure that the references are numbered consecutively.

1.2.- There are references that are not cited in the text: ref. 40, 41, 42, 45, 46, 47, 48, 49, 50, 51. They should be included in the text or deleted.

1.3.- There are references where outdated accession dates (2018 or 2020) are given. It is recommended that in references 20, 33, 35, 37, 38, 39 and 41 a confirmation is made that the links are still active and the date of the last access is updated.

1.4.- There are references in which the doi is included, but in others that should have it, it is not provided. It is recommended that the references be standardised, including the doi whenever it is available.

1.5.- There are references relating to some current topics that are obsolete. For example, reference 9 is from 1999 and refers to variability in data processing. With the emergence of a more fashionable thematic field such as Big Data, there is a lack of more up-to-date references. It is recommended to review the references and try to update the old ones with more modern sources.

2.- The paragraphs associated with lines 339/340 and 341-343 appear to be duplicated and provide the same information.

3.- In line 483 one line appears to be incomplete. It should be checked to see if part of the text has been inadvertently cut. I find it difficult to associate the context with what is indicated in that line.

4.- On line 490 the legend table 01 (Nomenclature) is repeated, when there is already a table 01 on page 8, associated with "Characteristics of the Soil Moisture Sensor".

5.- From lines 498 to 516 there is an unnecessary blank space that should be eliminated.

Author Response

The authors would like to thank the Reviewer for his valuable comments that helped to improve the quality and the clarity of the paper.

Reviewer 2 Report

1、  Agriculture 4.0 is a big concept. The example in this article is just a comparison of NDVI by satellite and drone. Why can it be called an experiment of Agriculture 4.0? In fact, the paper only discusses the scale of remote sensing as an experiment, but does not solve the problem. The article is more like a review, the content is not focused enough, and the innovation needs to be improved

2、  There is a problem with the L191 symbol; there is a writing error in the text of L608-609;

3、  The quality of Fig.4 is too poor to see clearly

4、  Is the GIS system made for this work? What are the important functions? The article does not introduce in detail

Author Response

The authors would like to thank the Reviewer for his valuable comments that helped to improve the quality and the clarity of the paper.
Please see the attachment.

Reviewer 3 Report

Experimentation on a study area for agriculture 4.0

Dear Authors

The basic science of this paper is conducted not in a good way and is of inappropriate standard.  The author and his team write this paper according to journal scope and modern trends. I am glad to review this paper but there is no novelty in this paper. I have seen many papers related to this topic and the study area has been published in well-reputed journals. If authors want to publish this study, they should provide some novelty or enhance the significance of the research. Moreover, the paper is not well-structured. I am going to recommend rejecting this paper at this stage. I hope the author will follow our comments and enhance their own study and resubmit again in this journal.  

Major comments

Title

•Title is very short and needs to revise title in a good form

Abstract

•Author tries to write in a better way but still, there are some mistakes. The author should follow the content of the abstract. The abstract is very rigorous.  The author should improve this part

Background

Context

Objectives

Material and methods

Results/findings

Novelty and purposes of this research

Abstract section is very small. Provide the background then start the objective of the research.

Provide some quantitative results instead of theocratical.

At the end of the abstract provide what is the significance of this research.

Introduction

No need to add heading 1.1 after the introduction part

Secondly, it lacks references. The author should add references in the 1.1 section

The authors should rewrite the Introduction section, extract research questions and useful information from the literature and point out the shortcomings of previous studies, thus leading to the article's environmental significance and purpose. Authors should particularly pay attention to the literature review which should be more critical. The authors should detail the methodological novelties with the vast amount of existing literature in this area. 

The author should clearly explain the main objectives of this study with a central hypothesis which is missing in the Introduction

In this introductory section, there are so many reference problems. The author should check all references accordingly to the context.

I found many type errors in the whole manuscript.

There is some typo error. The author should double-check the whole manuscript.

Objectives are not clear at the end of the introduction part

Remove heading 1.2 from the section.

The introduction section is very lengthy with a lack of significance in research

I believe the authors can demonstrate this and it should only take a couple of paragraphs in the introduction and discussion to show it but it is very important to do so if they wish to publish in an international journal.

2. Materials and Methods

2.1. UAVs, self-driving tractors, and data fusion 

The literature is out of references. Need to cite all these data.

There is continuity in the literature.

The author should revise the whole manuscript as per the journal scope and new trends in the scientific community.

I hope the authors will improve this study and resubmit it again in this journal.

Best Regards 

Author Response

(The authors gave the same response as above.)

Round 2

Reviewer 2 Report

The manuscript has been revised according to my comments

Author Response

The authors are grateful to the Reviewer for his insightful comments, which helped to improve the paper's quality and clarity.
In his 2nd round report he wrote 'The manuscript has been revised according to my comments'.
On the recommendation of another reviewer we corrected some imperfections and improved English language and style in the following lines:

  • Lines 17, 18, 19, 23-24, 26, 32, 34, 43, 62-63, 75, 94, 96, 97, 123, 140, 164, 176-177, 197, 202, 204, 255, 259, 261, 267, and 331. 

Of course, these line numbers are to be read in the MS Word file but not in the pdf, whose line numbers do not correspond to the editable file due to the activated 'Track Changes' function.

Reviewer 3 Report

Accept in present form

Author Response

The authors are grateful to the Reviewer for his insightful comments, which helped to improve the paper's quality and clarity.
In his 2nd round report he wrote 'Accept in present form' but for English language and style he left 'Moderate English changes required'. 
We have therefore corrected some imperfections and improved English language and style in the following lines:

  • Lines 17, 18, 19, 23-24, 26, 32, 34, 43, 62-63, 75, 94, 96, 97, 123, 140, 164, 176-177, 197, 202, 204, 255, 259, 261, 267, and 331. 

Of course, the line numbers are to be read in the MS Word file but not in the pdf, whose line numbers do not correspond to the editable file due to the activated 'Track Changes' function.